# DHT-Based Blockchain Dual-Sharding Storage Extension Mechanism

**Jindong Zhao** [1], **Donghong Zhang** [1,*], **Wenxuan Liu** [1], **Xiuqin Qiu** [1] **and Vladimir Brusic** [2]

1 School of Computer and Control Engineering, Yantai University, Yantai 264005, China
2 School of Computer Science, University of Nottingham Ningbo China, Ningbo 315100, China
* Correspondence: jansonzdh0730@s.ytu.edu.cn

**Abstract:** The expansion of blockchain storage has become a major problem limiting the application of blockchain. From the perspective of improving the scalability of blockchain storage, a DHT (distributed hash table)-based blockchain dual-sharding storage extension mechanism (DBDSM) is proposed. The nodes in the network are divided into $m$ DHT clusters. Each cluster includes $n$ nodes, and stores $1/m$ of the transaction data, and the nodes within each cluster store part of data allocated to that cluster. In this way, node storage pressure is alleviated. Furthermore, a hybrid query mechanism has been designed to achieve efficient querying of transaction data, without changing the original state data query. Simulation results showed without changing the original state data query, that the storage space consumed by the nodes is only $s/(m \times n)$ of that used in the traditional method; when the number of faulty nodes in the cluster does not exceed $s - 1$, the integrity of blockchain data can still be ensured. For transaction data queries, the average number of hops was 1.99, greatly improving query efficiency in the sharded state.

**Keywords:** blockchain; sharding; DHT; Kademlia; overlap storage





## 1. Introduction

Since the concept of peer-to-peer electronic cash transaction emerged in 2008 [1], blockchain technology and its underlying architecture have been applied in diverse areas including financial transactions, smart contracts, data management, storage, and communication [2]. Blockchain is a distributed ledger, maintained jointly by nodes over a distributed network. Nodes in the network maintain a local copy of the ledger by executing transactions authenticated by the consensus protocol. After more than 10 years of practical tests, blockchain technology has been recognized by many industries for the security of its transactions, data transparency, and the immutability of its transaction records. This has helped to enhance trust in direct electronic transactions without the need for trusted third-party intermediaries. However, the nodes in the blockchain network locally store multiple replicas of the entire blockchain, causing pressure on storage resources. Certain applications generate large quantities of transactional data, such as product traceability and scenarios from the Internet of Things (IoT), presenting challenges that cannot be adequately addressed by typical blockchain storage mechanisms [3]. Rapid growth of blockchain data creates huge storage pressure on the participating nodes, increasingly forcing nodes with limited storage resources to withdraw from the blockchain network, causing vulnerabilities in the stability of the system [4]. The expansion of blockchain storage capacity has become a major problem that limits the applicability and usability of blockchain technology [5]. Blockchain storage solutions that reduce infrastructure requirements are urgently needed.

Sharding has been used for traditional database capacity expansion by splitting large data collections across multiple servers [6]. The Elastico system [7], proposed in 2016, is an applied sharding protocol for improving blockchain efficiency [8–10]. Since then, various blockchain sharding solutions have been proposed for blockchain capacity expansion,

making sharding a mainstream blockchain expansion technology. Sharding helps not only to avoid erosion of the extent of decentralization within the system, but also supports the expansion of blockchain storage capacity and reduces the severity of scalability problems faced by blockchain applications [11]. Sharding divides the complete blockchain network into multiple separate fragments, each of which is maintained by a defined group of network nodes. All transaction processing and the storage of state data are completed within the fragment [12]. Multiple transactions generated on the blockchain, through parallel processing within each fragment, enable approximately linear improvement of the transaction processing speed. Meanwhile, transaction data are stored in different fragments to reduce the pressure of single-node storage.

Here we propose a DHT-based blockchain dual-sharding storage extension mechanism—DBDSM—that uses sharding to achieve scalable storage capacity within the blockchain. DBDSM has been designed to enable nodes to realize their basic functions and simultaneously reduce data storage requirements. A mixed query mechanism for transaction data in the fragmented state is deployed to improve the efficiency of the transaction data queries. The main focus of the DBDSM is to improve blockchain storage expansion capabilities.

The main contributions of this paper are as follows:

(1) We designed a DHT-based blockchain dual-sharding storage extension mechanism— DBDSM. The transaction data of the whole network are processed by the second fragmentation and stored in nodes across different clusters, providing security and availability of data while reducing transaction data storage requirements at individual nodes, thus realizing the expansion of the blockchain storage capacity.

(2) A hybrid query mechanism for transaction data in the fragmented state is proposed. On the premise of not changing the query of the blockchain state data, the query efficiency of transaction data in the fragmented state has been improved through the master-node caching mechanism and the routing lookup mechanism.

(3) The overlapping storage of fragmented data is allocated to the cluster by nodes within the cluster, guaranteeing the availability of fragmented data even when some of those nodes fail.

In Section 2 of this paper, we introduce some of the related work. The statement of the problem is presented in Section 3. Section 4 describes the dual-sharding mechanism. The theoretical analysis is presented in Section 5. The design of a simulation experiment and the results of the simulations are presented in Section 6. Section 7 summarizes the findings, offers conclusions, and discusses future work.

## 2. Related Work

### 2.1. DHT

A distributed hash table (DHT) is a distributed storage technology. In the DHT network, each node is responsible for storing part of the network's data and maintaining part of the node routing. Distributed storage of data and searches for specific resources can be realized without a centralized server.

The Kademlia protocol is an implementation of DHT technology. Unlike the ring network of the Chord protocol [13], the Kademlia protocol can map nodes in the network to a logical binary tree. When a node joins the network, the Kademlia protocol deploys the SHA-1 hash function to generate a 160-bit unique ID for each node, reflecting the node's unique identity. The protocol then converts the ID into binary storage, which uniquely determines the node's position in the logical binary tree. In the Kademlia protocol, the logical distance between nodes can be quantified by each node's ID value, where the XOR operation of two node ID values is used for determination of the distance. The smaller the operation value, the closer is the logical distance between the two nodes. In the DHT network, nodes can be dynamically adjusted without affecting the robustness of the system.

*2.2. Sharding Storage Protocol*

Sharding storage technology originated in the field of database storage. With the dramatic growth of data generated on the Internet, databases also need constant expansion to optimize the storage space of each node. Sharding technology has been applied to optimize storage availability by reducing the redundancy of database storage [14]. In the context of blockchain storage expansion, sharding storage is regarded as the most likely solution to address the bottleneck of blockchain storage capacity [15].

Elastico [7] was an early application of sharding that addressed the problem of managing blockchain expansion. Its core idea is transaction sharding. The nodes of the whole network are randomly divided into several trading committees. Each committee does not interfere with the others, and they verify their transactions in parallel, which allows linear growth of the blockchain performance. However, the Elastico model does not fragment the state data. Therefore, nodes in the network still need to store all the state data. Elastico is considered to have been the first public blockchain protocol based on sharding [16].

OmniLedger [17] improved on Elastico by offering a sharded blockchain design based on security and horizontal scaling. This solution solved several limitations that were manifest in Elastico. OmniLedger is a full sharded ledger in which each sharding node stores only part of the ledger data rather than the entire ledger data. The transaction verification process is entrusted to the cross-sharding consensus protocol, resulting in improved transaction processing efficiency and the reduction of storage consumption by single nodes. However, OmniLedger performs regular sharding and reorganization operations which cause an increased communication load during the process of data migration. OmniLedger performs better than Elastico in the trade-off between storage scaling, system security, and decentralization.

SSChain [18] is a public chain full sharding protocol without the data migration overhead. It adopts a two-layer architecture of root chain network and sharding network. The root chain network is responsible for verifying the blocks generated by each sharding, where each sharding network maintains part of the ledger data and processes transactions belonging to the same sharding. The market incentive mechanism can adjust the computing power distribution of the whole network to avoid double-spending attacks. SSChain supports transaction segmentation and state segmentation, reducing the storage consumption of nodes in the network. This protocol adopts a node incentive mechanism to allow nodes to freely join fragments. Therefore, nodes do not need to be reorganized regularly, eliminating the communication overhead caused by data migration.

ElasticChain [19] is a sharded storage method based on repetition ratio adjustment. Nodes use the storage ratio adjustment algorithm to store each part of the blockchain in shards to ensure data availability, thus reducing the storage consumption of the nodes. ElasticChain contains two types of chains: POR chain and P chain. The former records the node reliability certificate, the latter stores data. Each node can assume one of three different roles: user, storage, and verification. The verification nodes ensure that the fragmented data is always stored in a node with high reliability. This is achieved by verifying and recording the reliability of storage nodes on the POR chain, breaking the decentralized characteristics of the blockchain. ElasticChain involves two types of chains and three types of nodes, making it more complex than other storage solutions.

RapidChain [20] was the first public chain protocol based on sharding, and can provide complete sharding of communication, computation, and storage overheads of processing transactions without the assistance of a third party. However, for different states in the network, each committee is randomly stored. When a transaction is generated in the network, the initiator and the receiver of the transaction are not necessarily members of the same committee. They appear in two shards for cross-validation. The efficiency of this design is diminished in systems that have frequent communication and state exchanges.

The Monoxide model [21] includes asynchronous consensus zones for linear scaling of the blockchain. Nodes in the network are divided into multiple parallel zones. This method retains a cross-shard consensus problem. Meanwhile, to ensure the correctness of

cross-regional transactions, it proposes eventual atomicity, which can enhance the security of a single zone.

Xie et al. [22] proposed research on blockchain storage extension based on DHT (DBSE). Using the third-generation Kademlia protocol DHT technology, the nodes in the network are divided into several clusters. Each cluster stores the complete blockchain data, and a special node-mapping mechanism is used in the clusters to save the sharded data across different nodes, reducing the storage pressure on the nodes. A dynamic cluster reorganization mechanism is also included, but the data stored in each node in the network can never exceed 256 MB. With practically infinite recombination possibilities for the clusters, the number of sharding replicas is continuously reduced, diminishing the reliability of the system. The query efficiency of the transaction data in the sharding state has not been analyzed. In the DBSE model, each node stores complete account data and does not affect the verification of transactions in the blockchain.

In the progress of their research, Xing et al. [23] proposed scalable blockchain storage system models (SMBSS). Analysis of the Bitcoin UTXO model [24] and experimental validation indicated that nodes can independently verify more than 80% of the transactions on the blockchain by storing only the latest 3000 blocks on the chain. Based on this finding, it was determined that nodes in the UTXO-based blockchain network need only to store new block data generated within a specific time window to ensure the transaction validation function, and need to store only part of the old block data. The SMBSS model reduces consumption of node storage resources while ensuring independent participation of the nodes in the transaction validation. However, nodes must dynamically delete some of the latest blocks that have been saved, initially making the system more complicated to implement. The SMBSS model affects neither the decentralization of the network nor the availability of data.

The Meepo model [25] uses the concept of transaction sharding. It includes two processes: cross-epoch and cross-call. To handle multi-state dependencies in contract calls, it also includes a partial cross-call merging strategy, providing cross-contract flexibility. Meepo uses a replay epoch to ensure strict transaction atomicity. However, the nodes still need to store all data since the system is mainly used for transaction sharding but not for storage sharding.

## 3. Problem Statement

Blockchain platforms can be either public or consortium blockchains. Public blockchains are represented by Bitcoin and Ether, and the dominant consortium blockchain is Hyperledger Fabric. The cross-sharding consensus problem occurs when the sharding scheme is used for the extension of public blockchain platforms. In the Fabric consortium blockchain, the consensus process involves ordering nodes for sorting transactions without the participation of peer nodes. The numbers of nodes in consortium blockchains are relatively stable and change less frequently than in public blockchains. In this paper we report a novel storage scaling strategy using Fabric ledger.

### 3.1. Storage Model

Data in Hyperledger have four parts: world state, block index, ledger data, and historical state index. Among these, the world state and ledger data are the most important components used in Hyperledger, which together form the blockchain. The data-storage structure of Hyperledger consists of a state database, a file system, and a historical database. The Fabric ledger architecture is shown in Figure 1.

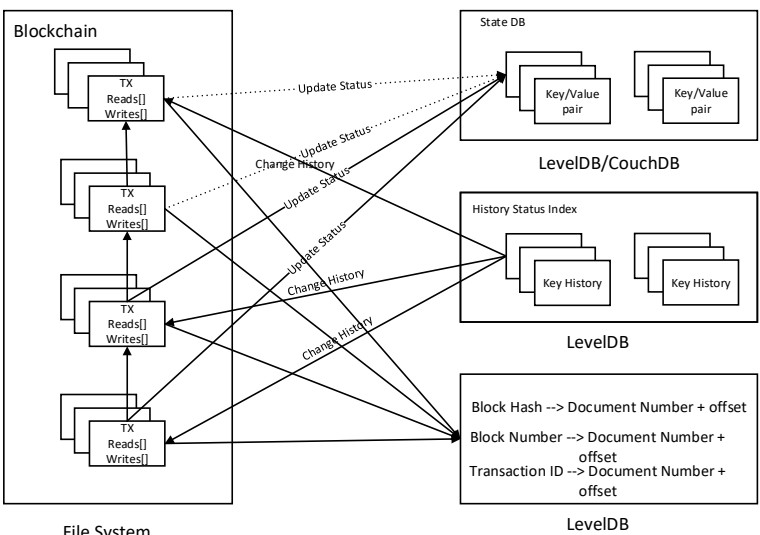

**Figure 1.** Fabric ledger storage.

The current values of all state data on the blockchain are stored in the state database, usually using databases such as LevelDB or CouchDB. The current value of an account status can be accessed directly through the world state, and the state data are stored as key-value pairs. The file system stores all the block data in the blockchain network. The blocks are linked with hash pointers, while the data information generated by the world state is recorded in the blocks. In Hyperledger, the peer nodes on the same channel store complete copies of the blockchain data. In the Hyperledger storage structure, the transaction data in the block are the main reason for the expansion of the blockchain storage capacity. Considering that the nodes require transaction verification functions, all nodes must store all the state data. Our solution uses block as the basic data unit to divide the data within the network.

*3.2. Storage Reliability*

In our solution, blockchain storage capacity expansion has been achieved by using sharding technology, where the nodes in the network abandon the original high-redundancy storage method and store only part of the blockchain data. However, this inversely affects the reliability of the system, and risk of data loss may occur when some nodes in the network fail. Therefore, it was important to ensure the reliability of the system used in the sharded storage capacity expansion scheme.

In the underlying design of the blockchain system, the traditional design model of p2p systems was adopted. The system reliability relationship is defined by:

$$r = \sum_{i=b}^{n} C_n^i p^i (1-p)^{(n-i)} \tag{1}$$

$$d = \frac{\log(1-r)}{\log(1-p)} \tag{2}$$

In Equations (1) and (2), $r$ denotes the reliability of the system, $p$ denotes the reliability of the nodes, $d$ denotes the number of replicas of the sharding, and $n$ is the total number of nodes. When node reliability is determined, the relationship between the system reliability and the number of replicas of the sharding can be calculated by Equation (2).

*3.3. Node Performance Evaluation Strategy*

In DHT clusters, to achieve efficient querying of transaction data in the sharding state, each cluster needs to elect a head node for a query, i.e., the master node. Therefore, it is

necessary to evaluate the performance of each node and then compare the stability values of all nodes in the cluster, before selecting the node with the largest stability value as the query master node. The calculation formula of the stability value $T(i)$ is:

$$
T(i) = \begin{cases} \frac{p}{1-p\cdot(m\%\frac{1}{p})}\cdot W, & i \in \mathrm{n} \\ 0, & otherwise \end{cases} \tag{3}
$$

The ratio of master nodes to total nodes in the network is designated by $p$. Because $1/p$ may appear as a non-integer, $1/p$ is rounded up and expressed as $1/p = \lceil 1/p \rceil$. The cluster label is denoted by $m$, while $n$ is the number of nodes in a cluster. Using Equation (3), we can calculate the performance of each node in the network. This equation provides the foundation for analysis of the query mechanism performance.

To analyze comprehensively the performance of the elected master nodes, we introduced control parameter $W$. The primary function of the master node is to achieve fast querying of historical data, so it is necessary to give priority to the node bandwidth parameter. The node with large bandwidth is preferentially elected as the master node. Equation (4) is used for calculating the parameter $W$:

$$
W = \alpha \cdot \frac{B_i(m)}{B_{avg}(m)} + \beta \cdot \frac{S_i(m)}{S_{avg}(m)} \tag{4}
$$

$\alpha$ and $\beta$ are weights where the relationship is expressed as: $\alpha + \beta = 1$. In specific applications, control of the weights of bandwidth and storage in the process of calculating stability values can be achieved by varying the values of the two weight factors $\alpha$ and $\beta$.

$B_i(m)$ denotes the bandwidth of the $i$th node in the $m$th cluster. $B_{avg}(m)$ denotes the average of the bandwidth values of all nodes in the $m$th cluster, calculated by Equation (5):

$$
B_{avg}(m) = \frac{\sum\limits_{i \in n} B_i(m)}{n} \tag{5}
$$

$S_i(m)$ denotes the storage capacity available at the $i$th node in the $m$th cluster. $S_{avg}(m)$ denotes the average of the storage capacity of all nodes in the $m$th cluster and $n$ denotes the total number of nodes in the cluster, calculated by Equation (6):

$$
S_{avg}(m) = \frac{\sum\limits_{i \in n} S_i(m)}{n} \tag{6}
$$

## 4. Dual-Sharding Mechanism

### 4.1. Cluster Division

Dual sharding uses the Kademlia protocol to divide the nodes in the network into DHT clusters. The Kademlia protocol generates a unique ID for each node based on the IP and Mac information of the nodes joined in the blockchain network. It uses the Sha1 algorithm (www.rfc-editor.org/rfc/rfc3174 (accessed on 16 July 2022)) to hash the unique information of the nodes, with an ID length of 160 bits. Therefore, $2^{160}$ nodes can theoretically exist in the network. By converting the ID value of each node into binary storage, the nodes in the network can be organized in a binary tree. All leaf nodes in the tree constitute the complete blockchain network, and the node position is uniquely determined by the shortest prefix of its ID value. We divide the peer nodes in the network into $m$ clusters, and each cluster consists of $n$ nodes. Each cluster needs only to store part of the data assigned to it by the blockchain network $(1/m)$, obviating the need to store all the data on each cluster. The nodes in the cluster overlap to store the data assigned to that cluster by the blockchain network, and each sharding stores $s$ ($s < n$) copies within the cluster. Therefore, the nodes in the cluster need to store $s/n$ of the data assigned to that cluster to ensure data security. Each node in the network needs to store all the state data.

The clusters are divided according to the order of the leaf nodes in the binary tree, from left to right. For example, 12 nodes are divided into four clusters of three nodes each (Figure 2).

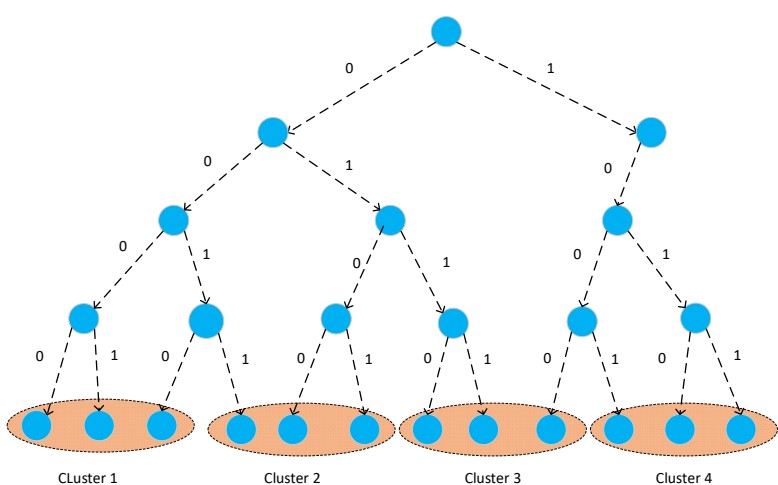

**Figure 2.** Cluster division.

If the number of nodes is not divisible by $n$, the number of nodes in the last cluster is less than $n$. Each node in this cluster needs to store all the data of the cluster until new nodes join the cluster, so that after the number of nodes reaches $n$, the data in the cluster is dynamically reorganized based on the set of overlapping shard groupings.

In our solution, the nodes in the blockchain network are divided into master nodes and ordinary nodes. Using Equation (3), the stability value of each node is calculated by considering the node network bandwidth, storage capacity, and other information. Based on the calculation results, AODV routing protocol [26] is applied to compare the stability value of each node in the cluster, and the node with the largest stability value is finally selected as the master node. A master node exists in each cluster. There is no difference between the master node and the normal nodes in terms of storage. Therefore, the storage mechanism of the blockchain nodes does not change. Each master node maintains a routing table, which stores the routing information of the remaining master nodes and the routing information of the common nodes in this cluster, to enable easy querying.

*4.2. XOR Mapping Mechanism*

We propose a shortest suffix sharding mapping mechanism, based on the Kademlia protocol. When a new block is packaged in the blockchain network, it generates a unique hash value, which we select as the unique ID of the new block and represent in the binary tree. The block ID value is subjected to shortest suffix XOR operation with the ID value of the nodes in its assigned cluster, The specific number of bits of the operation is $\lceil \log_2 n \rceil$, and the $s$ nodes associated with the operation from smallest to largest are selected as the nodes assigned for storage of this sharding. When some nodes in the network fail, the integrity of the data in the network can be maintained. The robustness of the system is ensured, even with the advantage of reduced redundancy of system storage.

Because our solution uses a dual-sharding mechanism, it is necessary to determine which cluster a block is assigned to after it has been generated. The scheme performs a specific operation based on block number $b$ and cluster number $i$, to determine whether or not the cluster keeps the block, with the following operation rules:

Define the operation $|$, for any non-zero integers x, y.

If x%y is not equal to 0, x | y = x%y.

Otherwise: x | y = y.

The initial value of block number $b$ is 1, and it increases sequentially, $b = 1, 2, 3, \ldots$

Figure 3 shows the process of sequentially assigning new blocks to the cluster and mapping them to the final node storage, with $m = 4$, $n = 3$, and $s = 2$ as an example.

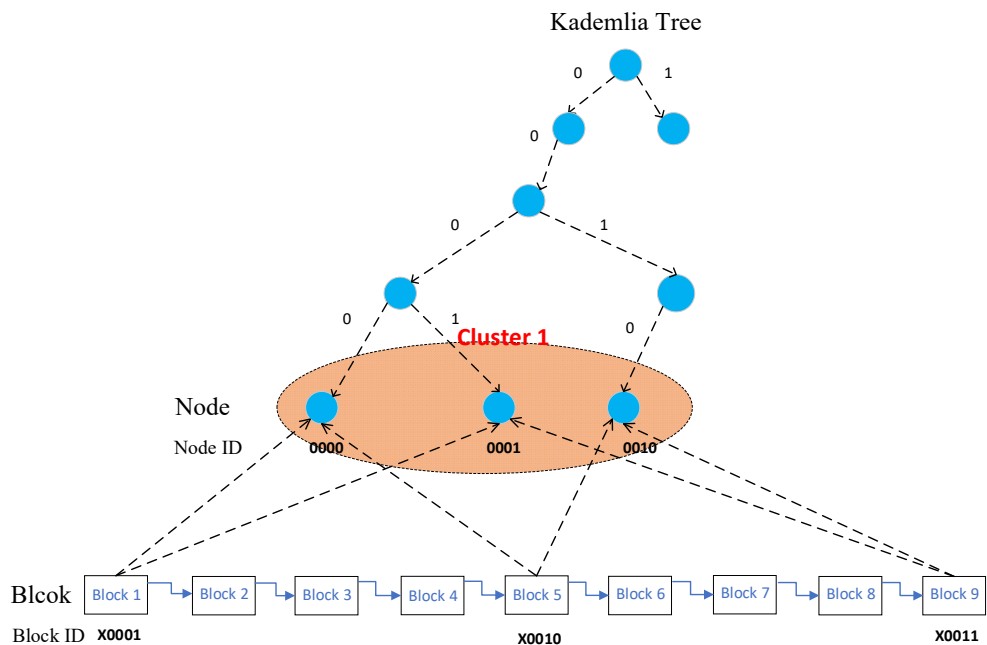

**Figure 3.** Data storage mechanism in cluster 1.

(1) The first generated block number is 1. The storage cluster location calculation formula yields 1|4 = 1, determining that the block should be stored in cluster 1. Then, according to the last-bit XOR mapping mechanism, the XOR operation is performed between the ID and the last $[\log_2 n]$ bit of the ID of the node in the cluster. The two nodes with the smaller operation values are selected as the storage nodes, and the final nodes stored in this block are determined to be nodes 0000 and 0001;

(2) The subsequently generated blocks numbered 2, 3, and 4 are calculated and stored in clusters 2, 3 and 4, respectively. The process of mapping blocks to nodes is the same as in (1).

(3) When the block number is 5, the storage location calculation formula yields 5|4 = 1, determining that the block is stored in cluster 1. According to the last-bit XOR mapping mechanism, the final nodes for the block storage are nodes 0000 and 0010.

(4) Continuing the process of (2) above, when the resulting block number is 9, the storage location calculation formula yields 9|4 = 1, determining that the block is stored in cluster 1. Then, according to the last-bit XOR mapping mechanism, the final nodes of the block storage are nodes 0001 and 0010.

*4.3. Hybrid Query Mechanism*

In a blockchain network, nodes adopt a kind of high-redundancy storage structure to store all of the blockchain data, including the state data and the transaction data. Therefore, the query process is divided into two types: one directly querying the state database to obtain data such as the asset information of an account; the other querying the transaction data, such as that generated by a user's asset transaction on the blockchain. When querying transaction information, it is necessary to traverse the complete local blockchain replica until the transaction information is acquired. In the storage design for the scheme presented here, the mechanism for storing all the state data in each node in the network is retained, and only the transaction data in the block is sharded. Therefore, querying the state data is consistent with the original Hyperledger query mechanism. However, when querying the

transaction data, since the sharded transaction data is stored in overlapping s nodes, it is necessary to send query requests to neighboring nodes in the network to traverse the query until the result is returned.

In the DBSE model, no query mechanism has been designed for the sharding state, so the query process adopts the original broadcast mechanism, which performs flooding lookup in nodes and sends query requests to neighboring nodes at random. If no query result is returned, the neighboring nodes continue to send query requests to their adjacent nodes until the query result is returned.

In the p2p network, there are a certain access patterns for data access. The access requests obey the zifp model distribution [27]. In this model, 10% of the nodes bear ~90% of access requests, and the remaining nodes bear ~10% of access requests. According to the principle of hot data access, data currently being accessed have a high probability of being accessed in a short period of time [28]. Based on the hot data access principle, we propose an efficient hybrid query mechanism to solve the problem of reduced query efficiency in the blockchain network sharded storage state. The query structure diagram is shown in Figure 4.

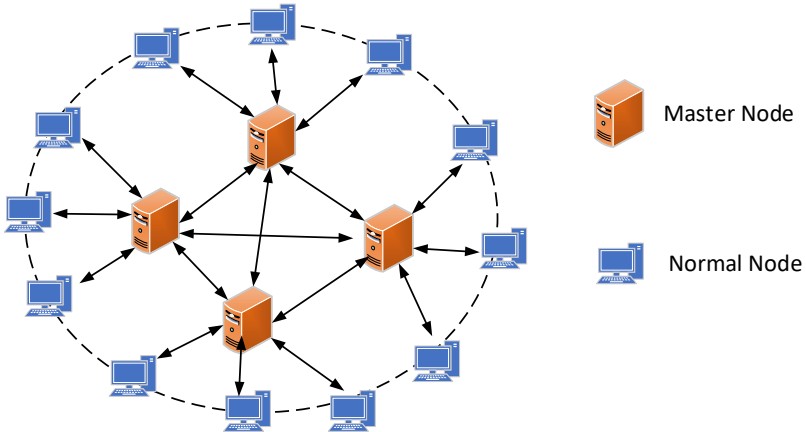

**Figure 4.** Query structure diagram.

(1)    Master node cache mechanism

After a user makes a transaction data query request, the query result is cached in the cache database of the master node of the cluster. When the user queries the same data again, the data is directly searched from the cache database of the master node of the cluster, which improves query efficiency. When cached data reaches 256 MB, the cached data is dynamically deleted in chronological order, which reduces the redundancy of node data storage and reduces the consumption of storage resources.

(2)    Route lookup mechanism

At the time of a transaction data query, there are master nodes and ordinary nodes in the network. Each master node in the network stores the routing information of the remaining master nodes, as well as the routing information of the adjacent ordinary nodes. The ordinary nodes need to store at least the routing information of the master nodes. When user sends a query request, if the required transaction data cannot be queried locally, the search moves to the adjacent master nodes through the routing information and gives priority to querying in all master nodes. If still no information is returned, it continues to query all ordinary nodes adjacent to the master nodes until the data is returned. Since storing routing information consumes very little node storage and has almost no storage impact, data lookup based on routing information greatly improves query efficiency without affecting storage redundancy.

The algorithm design of the hybrid query mechanism is shown in Algorithm 1.

---

**Algorithm 1** Transaction data query algorithm.

---

**Initialization:** $S = 0$
**Input:** node N, condition Ki//Ki means account key.
**if** N is ordinary node **then**
access master node
**end if**
**while** $S > 0$
**for** j ⟵ 0 to $S$
**if** get result **then**
return data
update cache data time
**end if**
**end for**
**end while**
Iterate through M local file system and M storage routing information, and query other master nodes and common nodes
return data
**if** cache < 256 MB **then**
Broadcast data to all master node caches
$S = S + 1$
**else**
Delete data that has not been accessed for a long time
Broadcast data to all master node caches
$S = S + 1$
**end if**
**Output:** transaction data

---

## 5. Theoretical Analysis

### 5.1. Storage Efficiency Analysis

Data stored in the blockchain are mainly state data and the transaction data. In this scheme, each node needs to store the state data to enable transaction verification. For the transaction data, the nodes in the cluster only need to store the part of data assigned to the cluster. Let $|H|$ denote the blockchain size, $|W|$ denote the state data size, $|S|$ denote the size of the master node cache data, $m$ is the number of clusters, $n$ is the number of nodes in a single cluster, and $s$ is the number of copies of the slice in the cluster.

In this scheme, a single cluster needs to keep $1/m \times |H|$ number of blocks, and the total amount of data stored by any ordinary node in the network is $s/(m \times n) \times |H| + |W|$. Therefore, compared with the original blockchain, the storage rate ($R_n$) of an ordinary node in the sharded state is:

$$R_n = \frac{\frac{s}{m \times n} \times |H| + |W|}{|H| + |W|} \tag{7}$$

The total amount of data stored by the master node in the network is $s/(m \times n) \times |H| + |W| + |S|$. Therefore, the storage rate ($R_n$) of the master node in the sharding state is:

$$R_n = \frac{\frac{s}{m \times n} \times |H| + |W| + |S|}{|H| + |W|} \tag{8}$$

When the transaction data storage capacity is much larger than the state data storage capacity, the storage space consumed by the common node tends to be $s/(m \times n) \times |B|$, see Equation (7). This scheme design supplies the master node with a dynamic adjustment mechanism to limit its cache capacity to no more than 256 MB. From Equation (8), the storage space of the master node tends to be $s/(m \times n) \times |B|$ when the transaction data storage capacity is much larger than either the state data storage capacity or the cache data storage capacity. When $s = n$, each node in the cluster stores all the slice data assigned to

that cluster, and the storage overhead is at its maximum; if $s = 1$, it means that a slice is stored on only one node in the network with no storage copy, and the storage overhead is minimal.

### 5.2. Query Efficiency Analysis

This scheme provides a hybrid query mechanism. When a node initiates a transaction data query request, the best case scenario is that the data requested by the node were requested a short time ago and are cached in the master node cache database. In such a case, the node only needs to query the neighboring master node cache database to obtain the requested data, and the whole query process requires only one or two hops. The worst case is when a node initiates a query request, the master node adjacent to the node is in a fault state and neither the cached data nor the routing information can be provided. The system can only use the flood query algorithm to query data on the nodes in the network, whereby the query process can reach up to six hops according to the six-degree separation theory. The specifics of query efficiency are described in the reports of the simulation experiments, in Section 6.

### 5.3. Security Analysis

Within a DHT cluster, the reliability of the system is mainly dependent on the storage policy and the data recovery policy. The storage policy refers to the redundancy method and the node selection mechanism is used when nodes store data. In this scheme, the number of replicas of each sharding is $s$. The larger the value of $s$, the more replicas of the sharding, the greater the storage space consumption of a single node, and the higher the system reliability; a smaller value of $s$ indicates fewer replicas of the sharding, less storage space consumption in a single node, and lower reliability of the system. The node selection mechanism adopts the last-bit XOR mapping method to distribute the blocks generated by the blockchain network evenly among the nodes in the network, to ensure reliability of data storage. The data recovery strategy means that the data maintain integrity even after failure of some nodes in the cluster. Suppose that among $n$ nodes, $x$ nodes are randomly selected as the failed nodes, while the remaining $n - x$ nodes continue to keep all the blocks of the cluster intact. For a particular $n$, the larger the value of $x$, the higher the reliability of the system, and the redundancy in the system is greater at that time.

When $n$ is a fixed value, the system reliability is related to $s$. When $s = 1$, each block is stored in only one node. At this time, $x$ must be 0 and no node is allowed to fail, or such a failure will lead to unrecoverable blockchain data; when $s = n$, at this time $x$ has a maximum value of $n - 1$, and even if $n - 1$ nodes fail, the full blockchain data can be recovered.

## 6. Experiment

### 6.1. Experiment Setup

The experimental machine was configured with an Intel(R) Core (TM) i7-8565U CPU @ 1.80 GHz 1.99 GHz and 64 G RAM, and the operating system was CentOS 8. Python language was employed to complete the experiment, and Python-igraph was selected to create a Watts–Strogatz network, simulating nodes in the blockchain network.

### 6.2. Experimental Results

Simulation experiments were carried out on the state data, transaction data, query efficiency, and the blockchain to verify the storage efficiency, query efficiency, and data availability provided by this solution.

During the experiment, a small-world network with 100 nodes (*dim* = 3, *size* = 5, *nei* = 1, *p* = 0.1) was created with the use of the igraph library in Python. Based on the cluster division method designed in our system, the nodes in the network were divided into four clusters. At this time, $m = 4$, $n = 25$, and the number of sharding replicas $s$ was set to $s = \lceil 2n/3 \rceil$, i.e., two

thirds of the total number of nodes in the cluster rounded upward to obtain the number of replicas of the sharding. The small-world network model is shown in Figure 5.

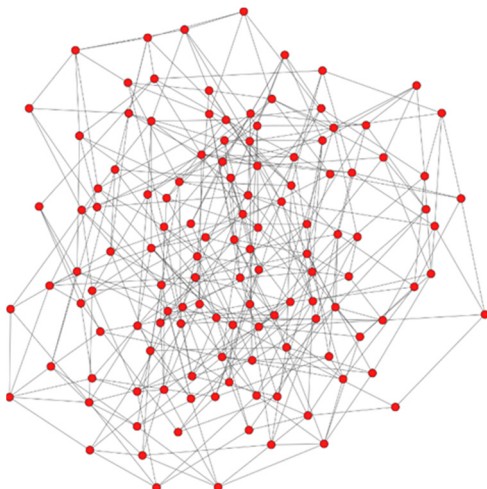

**Figure 5.** Small-world network model.

Experiment 1: Single-node storage consumption.

In the experiment, we set the size of a world's state record to $S_w$, transaction data to $S_t$, and a block to $S_b$. For simplicity of the experiments, the block header data were ignored. Transactions were simulated between different world states, and the transaction data were stored. When the transaction data reached the threshold of the specified block size, a new block was generated and broadcasted, and the block was stored on a peer node in the network according to the design of this scheme.

The network was initialized according to the Fabric default values, setting the $S_b$ value to 512k, while each block contained 10 transaction data, bringing the $S_t$ value to 51.2k. The variable $N_t$ indicates the total number of transactions generated in the network, and $N_w$ indicates the total number of world states generated in the network. The experimental results of the simulated storage consumption within a single node are shown in Figure 6.

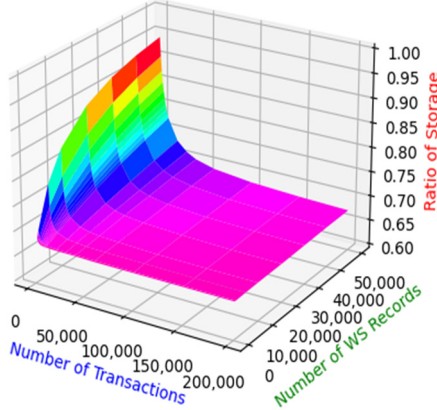

**Figure 6.** Single-node storage consumption.

By analyzing the experimental results in Figure 6, it can be concluded that:

(1) When $N_t$ was less than or equal to $N_w$ in the network (one world state corresponding to at most one transaction), there was no significant change in the storage space consumed by DBDSM nodes compared with the Fabric node data storage. When $N_t$ dimin-

ished, as $N_w$ increased, the DBDSM node storage consumption was closer to the original Fabric node storage consumption.

(2) When $N_t$ in the network was larger than $N_w$, with one world state corresponding to multiple transactions, DBDSM node storage consumption appeared significantly reduced compared with the Fabric node data storage. When $N_t$ was much larger than $N_w$, compared with the transaction data, the impact of state data on node storage consumption can be neglected. At this time, the storage consumption of DBDSM nodes was smaller compared with the Fabric nodes, at 66% of the storage consumption of Fabric nodes.

From the experimental results in Figure 7, it can be concluded that:

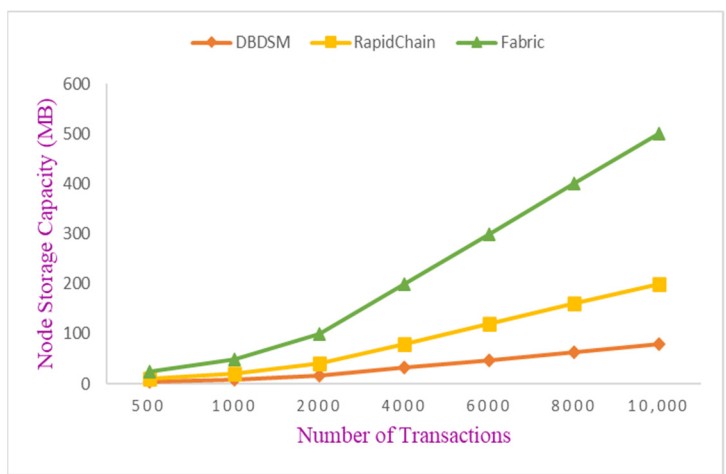

**Figure 7.** Storage space occupied by DBDSM, RapidChain and Fabric.

(1) When the number of transactions generated in the network was small, the data generated were also small, and the quantities of data stored by the nodes in the DBDSM model, RapidChain model, and the Fabric model did not differ greatly. Because each scheme stores state data, when the transaction data were small, the main impact on storage was restricted to the state data.

(2) With the increase in the number of transactions generated in the network, compared with Fabric, the DBDSM model and the RapidChain model tended to flatten the increment of node storage data, because both the DBDSM model and the RapidChain model use a sharding scheme, reducing the number of single nodes and the amount of data stored by each node.

(3) When the number of transactions generated in the network continued to increase, the growth rate of data storage in the DBDSM model nodes was slower than in RapidChain. We can conclude that the DBDSM model has good scalability performance.

Experiment 2: Query efficiency.

Each node in this scheme stores the state data, so the query for the world state is consistent with that of Fabric. For the query of transaction data, based on the original gossip protocol query, our scheme proposes a hybrid query mechanism for efficient querying through the master node caching mechanism and routing mechanism, using a hybrid query algorithm. In this experiment, we used data results generated from Experiment 1. A node was randomly selected to query the specified transaction. If the selected node was the master node, the transaction was first sought in the cache database, and if no transaction was returned, the transaction continued to be queried from the node's stored blocks, and if the desired transaction was obtained, the network overhead of the query was recorded as one. Otherwise, the network overhead was recorded as one, according to the node's other master and the routing information of ordinary nodes. If the selected node was not the master node, the lookup was performed locally, and if results were returned, then the communication overhead was given a value of one. Otherwise, the master node and

ordinary nodes were queried through the routing information, and the communication overhead increased by one point for each routing information lookup. Following this process, 100 transactions and 1000 transactions were considered, and 1000 transaction data queries were performed for each set. The experimental results are shown in Figure 8 and the statistical results of different models are shown in Table 1.

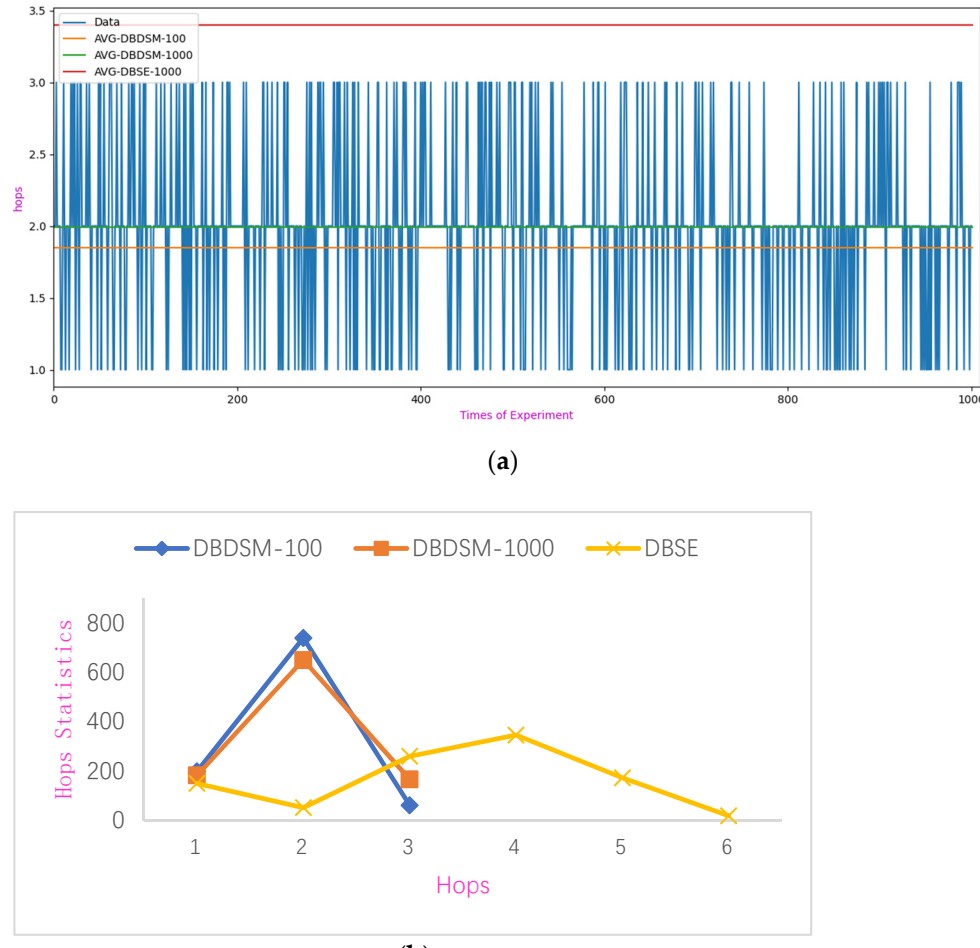

(a)

(b)

**Figure 8.** Query efficiency result statistics. (**a**) Transaction data query communication overhead. (**b**) Query result statistics.

**Table 1.** The statistical results of different models.

| Hops<br>Model Frequency | 1 | 2 | 3 | 4 | 5 | 6 |
|---|---|---|---|---|---|---|
| DBSE | 150 | 52 | 260 | 346 | 173 | 19 |
| DBDSM-100 | 199 | 739 | 62 | 0 | 0 | 0 |
| DBDSM-1000 | 183 | 650 | 167 | 0 | 0 | 0 |

By analyzing the experimental results in Figure 8, it can be concluded that:

(1) Under the same query conditions, the network communication overhead required to perform 1000 random transaction queries in this scheme was usually within two and not more than three hops. The DBSE network communication overhead was usually three or four hops and not more than six hops. The original DBS scheme adopted a broadcast query mechanism, where the data query process was flooded, so the query efficiency was lower. In contrast, the master route lookup mechanism used by DBDSM in this scheme can

quickly locate the required transaction data based on the route information maintained by the nodes when performing transaction data queries, greatly improving the query efficiency in the sharding state.

(2) A comparison of the query range for different transactions shown in Figure 8b reveals that when 1000 queries were made randomly for 100 transactions, the communication overhead was usually two hops or one, with an average communication overhead of 1.85; when 1000 queries were made randomly for 1000 transactions, the communication overhead was usually two hops or one, with an average communication overhead of 1.99, meaning that compared with 100 transactions, the query efficiency was slightly lower. Our scheme uses a transaction data caching mechanism, where a node performs multiple query operations on the same transaction data, and the master node in the network caches the transaction after the first query operation. In the subsequent multiple queries, a node only needs to access the adjacent master node and query its cache database. The cache database uses a dynamic deletion mechanism, greatly improving the efficiency of data queries in the sharding state without affecting the storage redundancy of master nodes.

Experiment 3: System reliability and node storage rate analysis.

Assuming that the average availability of nodes in the blockchain network is 0.7, the probability of node failure is 0.3. When $n = 25$, denoting the average number of copies of data allocated in the cluster, $b$ denotes the number of nodes required to recover the complete data. When $b$ takes different values, the system reliability results are calculated from Equation (1).

By analyzing the experimental results in Figure 9, it can be concluded that:

(1) When there were two overlapping replicas of the sharding in the blockchain network, the reliability of the system was at its worst; as the number of overlapping replicas of the sharding increased, the reliability of the system increased gradually, and when the number of replicas reached 16, the reliability of the system was 99.9%. Therefore, the system was reliable when the number of overlapping replicas of the sharding reached 16.

(2) The comparisons in Figure 8a,b show that, unlike the system reliability, the node storage rate increased linearly with an increasing number of overlapping sharding replicas. When the number of replicas was two, the storage rate of the nodes was the lowest and the system reliability was at its worst; when the system reliability reached 99.9%, the corresponding node storage rate was 64%. Therefore, the system's node storage consumption was effectively reduced while its reliability was maintained.

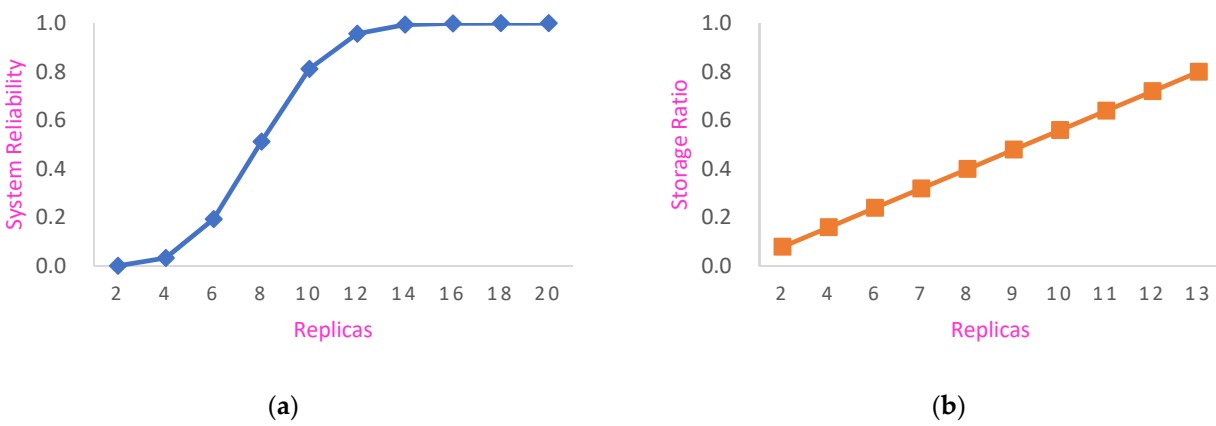

**Figure 9.** System reliability and node storage ratio. (**a**) System reliability. (**b**) Node storage ratio.

## 7. Conclusions

Blockchain technology has developed rapidly in the past few years, and blockchain technology will expand to more fields of application in future. Therefore, blockchain scalability is a problem that must be considered and solved. In this paper, we propose a DHT-based

blockchain dual-sharding storage extension mechanism. Our solution uses DHT technology and sharding technology to realize sharding storage of blockchain data and at the same time reduce the storage consumption of nodes. Through the XOR operation mechanism, sharding is evenly mapped to different nodes in the cluster for overlapping storage, ensuring the safety and reliability of the sliced data while effectively solving the blockchain storage scaling problem. We propose a hybrid query mechanism for transaction data under the sharding storage state, to improve the query efficiency of transaction data under the sharding state with almost no impact on node storage consumption. Simulated experimental results showed that the DBDSM model can meet the blockchain scalability requirements.

This scheme is currently applicable to licensed blockchains. For non-licensed blockchains, such as Bitcoin and Ether systems, this scheme is not applicable due to the cross-partition consensus involved between different partitions. Future research will involve further optimization and improvements based on this scheme, so that it can be applied to non-licensed blockchain systems. Optimization and scalability of blockchain solutions are key issues for applications that deal with big data. There are significant implications for growing areas of research such as the collection, analysis, storage, and use of big data collected by drones, smart agriculture, the security of big data systems, medical applications, machine learning from IoT data, and applications across smart cities [29,30].

Potential applications of our solution include blockchain applications in the IoT environment where large data streaming results need be transferred securely, safely, and with good scalability properties. These may include industrial IoT blockchains, mobile computing applications, social data storage, collaborative computing, and smart home computing, among others [31–35].

**Author Contributions:** Conceptualization, J.Z., D.Z. and W.L.; methodology, J.Z. and D.Z.; software, D.Z., W.L. and X.Q.; data curation, D.Z. and W.L.; validation, J.Z., X.Q. and V.B.; writing—original draft, J.Z. and D.Z.; writing—review and editing, X.Q. and V.B.; supervision, V.B. All authors have read and agreed to the published version of the manuscript.

**Funding:** This work was financially supported by the National Natural Science Foundation of China (No. 61972360), and the Natural Science Foundation of Shandong Province, China (No. ZR2020MF148).

**Institutional Review Board Statement:** Not applicable.

**Informed Consent Statement:** Not applicable.

**Data Availability Statement:** Not applicable.

**Acknowledgments:** The authors of this paper acknowledge Xiang Zhang from the University of Nottingham Ningbo China for making improvements to the language and expression of this paper.

**Conflicts of Interest:** The authors declare no conflict of interest.

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
