# Peer review of "DHT-Based Blockchain Dual-Sharding Storage Extension Mechanism"

_applsci, doi:10.3390/app12199635_

Round 1
Reviewer 1 Report
I suggest to expand cited references, improve the presentation of the results and expend the conclusions of the results.
Minor language check needed.
Author Response
1.Comment: I suggest to expand cited references, improve the presentation of the results and expend the conclusions of the results.
1.Reply: Thank you for your comments. We have expanded cited references in section 2.2, References 18,19,23 were added. We have also used Table 1 to improve the presentation of the experiment 2 results in section 6.2 and marked in red in the revised version.
2.Comment: Minor language check needed.
2.Reply: Thank you for your suggestions. We have checked language carefully and fixed some problems. Thank you again for your careful check.
Reviewer 2 Report
The article contains a proposal for solving problems that are related to the scalability of blockchain storage. The subject in this paper is a proposal for a storage scaling strategy for a Consortium blockchain Fabric. The authors have done a literature review of existing sharding solutions and, an overview of the challenges and theoretical analysis. They give mathematical formulations of the reliability and performance strategy of the chain nodes. Section 6 consist presented the dual shredding mechanism, hybrid query mechanism, caching and routing mechanisms, storage and query performance analysis, and the simulation experiment is performed. The research concepts of the proposed mechanisms are validated with experimental results and the application of mathematical formulations. The research and results are presented correctly and in a style accepted by science.
There is an unclear sentence or maybe I don't understand. “However, with its continuous application in various scenarios, such as product traceability and other scenarios that generate a large number of transaction data information”
Author Response
1.Comment: The article contains a proposal for solving problems that are related to the scalability of blockchain storage. The subject in this paper is a proposal for a storage scaling strategy for a Consortium blockchain Fabric. The authors have done a literature review of existing sharding solutions and, an overview of the challenges and theoretical analysis. They give mathematical formulations of the reliability and performance strategy of the chain nodes. Section 6 consist presented the dual shredding mechanism, hybrid query mechanism, caching and routing mechanisms, storage and query performance analysis, and the simulation experiment is performed. The research concepts of the proposed mechanisms are validated with experimental results and the application of mathematical formulations. The research and results are presented correctly and in a style accepted by science.
There is an unclear sentence or maybe I don't understand. “However, with its continuous application in various scenarios, such as product traceability and other scenarios that generate a large number of transaction data information”.
1.Reply:Thank you for your comments. Because in the blockchain, data is mainly divided into two parts, transaction data and state data. However, transaction data is the main reason for the dramatic growth of blockchain data. When a large amount of transaction data is generated in the blockchain network, it will bring huge storage pressure to the participating nodes.
Reviewer 3 Report
1. In this paper it is proposed a new mechanism concerning the expansion the block chain storage. They authors have to examine the energy effect of this mechanism since the cost of energy is important factor.
2. The proposed technique seems to expand the storage mechanism. However the new mechanism should be compared with the existing one and examples should be given.
Author Response
1.Comment: In this paper it is proposed a new mechanism concerning the expansion the block chain storage. They authors have to examine the energy effect of this mechanism since the cost of energy is important factor.
1.Reply: Thank you for your comments. In this schema, we have not changed the original blockchain consensus mechanism, only the way of data storage, without adding additional energy costs. We have changed to the way we query, which will slightly reduce the efficiency of the query, but the impact on energy consumption is negligible.
2.Comment: The proposed technique seems to expand the storage mechanism. However the new mechanism should be compared with the existing one and examples should be given.
2.Reply: Thank you for your comments. We have added a comparative experiment between our mechanism and the Rapidchain mechanism in section 6.2. Analysis of experimental results is marked in red in the revised version.
Reviewer 4 Report
This work presents a solution to mitigate the large-ledger issue of the prevalent Blockchain architectures.
The paper overall, as well as the clustering solution it proposes are in the right direction.
The flow of the paper is fluent, and I really enjoyed going through the presented literature review. The work and the conception are nicely given.
I suggest it is accepted for publication.
Some minor significance modifications-suggestions to the authors.
To the best of my reading, the sharding-storage solution presented considers private (community) blockchain setup. To the best of my understanding, the proposed process for the introduction of new nodes goes through the “Kademlia protocol”, which is run by a “cluster-head” node, which inevitably becomes a single point of failure for the cluster, imposing the necessity for consensus achievement both during the “picking” of the master, as well as in the case of “contradicting” query results.
To the best of my understanding, it did not became clear to me if the consensus process of the proposed architecture (both during the “write” and during the “verification” process) requires the invocation of the “whole realm” or it is committed in a “single cluster”. Perhaps this could be highlighted more by the authors.
In my opinion, a paragraph on how the solution deals with consensus in the light of disputes/contradictions would add value to the paper.
Still, this became apparent to me for the storage process.
I found it hard to follow the annotations/symbols in the equations. I would suggest the authors to add brief explanatory text below each equation or an overall “Definitions” paragraph or table to make understanding easier.
Minor expression issues were detected throughout the text, as in lines 98, 129, 141.
Author Response
1.Comment: To the best of my reading, the sharding-storage solution presented considers private (community) blockchain setup. To the best of my understanding, the proposed process for the introduction of new nodes goes through the “Kademlia protocol”, which is run by a “cluster-head” node, which inevitably becomes a single point of failure for the cluster, imposing the necessity for consensus achievement both during the “picking” of the master, as well as in the case of “contradicting” query results.
1.Reply: Thank you for your comments. The Kademlia protocol is a p2p overlay network transmission protocol, usually operating in a p2p network. Especially used in blockchain technology, such as the Ethereum system. Kad establishes a new DHT topology through a unique XOR algorithm (XOR) as the basis for distance metric. In the Kad network, all nodes are regarded as the leaves of a binary tree, and the position of each node is uniquely determined by the shortest prefix of its ID value. Therefore, when the Kademlia protocol is used to build a DHT cluster in this schema, There is no cluster head node problem, so there is no single point of failure.
2.Comment: To the best of my understanding, it did not became clear to me if the consensus process of the proposed architecture (both during the “write” and during the “verification” process) requires the invocation of the “whole realm” or it is committed in a “single cluster”. Perhaps this could be highlighted more by the authors. In my opinion, a paragraph on how the solution deals with consensus in the light of disputes/contradictions would add value to the paper. Still, this became apparent to me for the storage process.
2.Reply: Thank you for your comments. Our schema performs sharding storage of transaction data based on the consortium blockchain fabric, and does not change the original consensus mechanism. In fabric, the consensus process is completed by the order node. Therefore, there is no need to consider the consensus reached by the entire network or a single cluster.
3.Comment: I found it hard to follow the annotations/symbols in the equations. I would suggest the authors to add brief explanatory text below each equation or an overall “Definitions” paragraph or table to make understanding easier.
3.Reply: Thank you for your suggestions. We have added brief explanatory text below each equation and marked them in red in the revised version.
4.Comment: Minor expression issues were detected throughout the text, as in lines 98, 129, 141.
4.Reply: Thank you for your checks. We have corrected all expression issues as you suggested and marked them in red in the revised version . Thank you again for your careful check.
Reviewer 5 Report
1. The paper is written for readers all over the world, not just for China. Therefore, phrases like "at home and abroad" should not be used.
2. The paper describes the relevance of the study very poorly. In my opinion, it is necessary to describe more specifically what caused the need for it.
3. The goal and plan of the study are poorly formulated in the article, which makes it difficult to understand the logic of its conduct.
4. It is advisable to summarize the results of the comparison of the analyzed systems in a table, which will simplify their understanding.
5. It is not entirely clear why the study, the results of which are presented in Figure 8, was conducted. The graphs show that they are determined by the simplest mathematical formulas. Therefore, it would be advisable to use a mathematical (analytical) approach to solving this problem.
Author Response
1.Comment: The paper is written for readers all over the world, not just for China. Therefore, phrases like "at home and abroad" should not be used.
1.Reply: Thank you for your suggestions. We have deleted phrases like "at home and abroad". Thank you again for your careful check.
2.Comment: The paper describes the relevance of the study very poorly. In my opinion, it is necessary to describe more specifically what caused the need for it.
2.Reply: Thank you for your comments. We have described the reasons for the need in detail and marked them in red in the revised version. There are two main reasons: the node storage pressure is excessive and waste of storage resources.
3.Comment: The goal and plan of the study are poorly formulated in the article, which makes it difficult to understand the logic of its conduct.
3.Reply: Thank you for your comments. We have formulated goal and plan of the study in section 1 and marked them in red in the revised version.
4.Comment: It is advisable to summarize the results of the comparison of the analyzed systems in a table, which will simplify their understanding.
4.Reply: Thank you for your suggestions. We have summarized the results of the comparison of the analyzed systems as you suggested. The results are showed in Table 1 in section 6.2 in the revised version.
5.Comment: It is not entirely clear why the study, the results of which are presented in Figure 8, was conducted. The graphs show that they are determined by the simplest mathematical formulas. Therefore, it would be advisable to use a mathematical (analytical) approach to solving this problem.
5.Reply: Thank you for your comments. Figure 8 shows the results obtained by corresponding calculation based on the system reliability calculation formula in P2P network and the nodes used in the experiment of this scheme. The calculation results need to be combined with simulation experiments to better verify the reliability of the scheme.